# Management of Delta Hepatitis 45 Years after the Discovery of HDV

**DOI:** 10.3390/jcm11061587

**Published:** 2022-03-13

**Authors:** Stefano Brillanti

**Affiliations:** Department of Medicine, Surgery and Neuroscience, University of Siena, 53100 Siena, Italy; stefano.brillanti@unisi.it

**Keywords:** hepatitis Delta, HDV, infection, interferon, bulevirtide, cirrhosis, treatment, therapy, myrcludex-B

## Abstract

In 1977 the viral Delta agent was discovered and subsequently characterized as the hepatitis Delta virus (HDV). HDV infection is associated with HBV infection since the defective HDV needs HBV to infect and replicate in the liver. Even if not a frequent cause of chronic liver disease, HDV infection is responsible for an aggressive progression of hepatitis towards advanced liver disease. At present, no FDA approved treatment exists for this specific form of hepatitis. Interferon alfa has been recommended as off-label therapy by major scientific societies (AASLD, EASL and APASL) and has proved effective in about one quarter of patients. In recent years, new therapeutic approaches have been studied, and EMA has approved a new drug (bulevirtide) for Delta hepatitis. In this review, we encompass the 45-year journey of managing Delta hepatitis and address the most recent developments in treating this severe and aggressive liver disease.

## 1. Introduction

Hepatitis Delta Virus (HDV) is the cause of hepatitis D, a relatively rare but aggressive form of viral hepatitis developing in patients co-infected with hepatitis B virus (HBV). Probably vastly underestimated in HBsAg carriers, the lack of standardized virological methods to diagnose and monitor the infection and effective therapy to treat the liver disease have conditioned the progress in research and clinical management. However, after almost half a century from the initial discovery of the Delta agent, recent developments in understanding and targeting the therapeutic efforts have opened new hope for patients with hepatitis D. In this narrative review, we try to summarize and focus on the new opportunities in the management of Delta hepatitis.

## 2. When All of This Started

Hepatitis Delta virus (HDV) is a defective RNA viral agent associated with HBV infection. In 1977, in Turin, Italy, Mario Rizzetto first detected by immunofluorescence a new antigen, named Delta, in the liver cell nuclei of patients with HBsAg positive chronic liver disease. Corresponding circulating antibodies were similarly found in the serum of chronic HBsAg carriers, especially those with liver damage [1].

However, it took a few years to understand that the new antigen was not secreted by HBV but represented an independent agent transmitted by superinfection or coinfection of HBsAg carriers. The new agent was characterized as a defective RNA pathogen dependent for replication and infection on helper functions provided by HBV. The virion is a 35–37 nm particle, with the Delta antigen and the RNA genome within a coat made by the HBsAg lipoprotein [2].

The prevalence of HDV infection is limited to individuals with HBV infection, as coinfection with HBV of normal subjects or superinfection of HBsAg carriers. Parenteral transmission is the route of propagating infection from one individual to another [3].

Since the 1980s, it was clear that HDV infection had high pathogenic potential, inducing hepatitis in all infected subjects. Hepatitis may be acute or chronic, the former more frequently associated with coinfection, and the latter mainly occurring after superinfection.

The HBsAg carrier state represents the ideal background for infection and replication of HDV, and the subsequent chronic hepatitis is generally active and severe, progressing to more advanced liver disease within a few years [4].

All of this has been known for decades, and we are still waiting for significant progress in the clinical management of Delta hepatitis.

## 3. How to Treat Patients with Chronic Delta Hepatitis

Because of the aggressive progression of Delta hepatitis towards liver cirrhosis and hepatocellular carcinoma (HCC), therapy has the primary goals of decreasing the development of severe outcomes, the need for liver transplantation and, ultimately, the patient’s death by liver-related causes. Intermediate surrogate goals are the normalization of transaminases, loss of circulating HDV RNA and, ultimately, HBsAg seroconversion.

Unfortunately, treatment of Delta hepatitis faces significant challenges: (1) patients often present a difficult to treat advanced liver disease, (2) transaminase flares are frequent on treatment or treatment withdrawal, (3) suppression of HDV replication may be associated with HBV flares, (4) lack of standardization of testing and response criteria, (5) association with autoimmune liver disease is relatively common and may hamper the use of interferon or other immunomodulating drugs, and finally, (6) HBV elimination should be the ultimate goal, but unfortunately, it is generally not achieved.

These challenges are partially due to the absence of new therapeutic choices and the suboptimal results obtained with available therapies, characterized by poor response rates and limited efficacy [3].

Because of the uncertainty of the long-term beneficial effects of HDV suppression and the difficulties in reaching HBsAg clearance, intermediate surrogate markers of treatment efficacy have been proposed and accepted: a >2 log drop in serum HDV RNA levels combined with aminotransferase (ALT) normalization have been accepted as a measure of initial treatment efficacy in clinical trials. In recent years, these intermediate and surrogate therapeutic endpoints have allowed testing the efficacy of new experimental drugs in HDV infection and HDV-related hepatitis [5].

## 4. The Role of Interferon and Nucleotide/Nucleoside (NUC) Analogues

In 1994, Patricia Farci and co-workers first demonstrated the administration of recombinant interferon alfa-2a (IFN-a2a), given at the dose of 9 MU subcutaneously three times a week for 48 weeks, induced ALT normalization as well as HDV-RNA loss in 71% of patients during treatment. In addition, a combined biochemical and virological response was achieved in 50% of treated patients but unfortunately was maintained in only 21% of them during the weeks after the end of therapy [6]. Assessing these results after ten years, the same authors demonstrated that patients treated with IFN-a2a (9 MU three times weekly) had a significantly better long-term survival and a more profound HDV replication inhibition than those treated with a lower dose of IFN-a2a or untreated. Unfortunately, HDV viremia relapse was found in all treated patients using sensitive PCR testing. Finally and importantly, paired liver biopsies showed a significant and lasted regression of liver fibrosis years after IFN-a2a therapy [7].

The advent of pegylated-interferon alfa (Peg-IFN) allowed different groups to test the efficacy of this new formulation of interferon in patients with chronic Delta infection. Unfortunately, Peg-IFN did not show significant advantages in controlling HDV infection: the loss of serum HDV-RNA after the end of therapy was not significantly improved in comparison with recombinant IFN-a2a, with positive results ranging between 23 and 43% (mean 25%) even after a prolonged therapy course of 24 months [8].

HDV is a defective agent needing HBV infection to replicate. For this reason, drugs able to inhibit and suppress HBV replication could be beneficial in treating Delta hepatitis. NUCs (entecavir and tenofovir) represent the standard treatment in patients with chronic hepatitis B, able to suppress HBV replication. Since 1999, different NUCs have also been used in patients with chronic Delta hepatitis. Unfortunately, NUC monotherapy or in combination with Peg-IFN has not demonstrated any significant, additive or long-lasting effect in controlling HDV infection [9,10,11].

At present, over 25 years of experience in the treatment of Delta hepatitis can be summarized as follows: recombinant or pegylated interferon alfa therapy can inhibit HDV replication and reduce liver inflammation and disease progression in 20–25% of patients; anti-HBV NUC monotherapy or addition is not beneficial, but may prevent HBV flares; interferon therapy is contraindicated in more advanced liver disease, has significant side effects, and may induce associated autoimmune hepatitis. For all these reasons, there is an urgent need for more manageable and effective HDV suppressive therapies.

## 5. New Drugs for Delta Hepatitis

Different approaches have been investigated to interfere and suppress HDV replication during the last years. As a result, three significant therapeutic mechanisms have been identified: (1) inhibition of HDV prenylation, (2) inhibition of HBsAg release and (3) inhibition of cell entry [12] (Figure 1).

Prenylation involves the covalent addition of a farnesyl or geranylgeranyl isoprenoid molecule to a conserved cysteine residue at or near the C-terminus of a protein. This link promotes membrane interactions with the prenylated protein since the isoprenoid chain is hydrophobic. Lonafarnib, a farnesyltransferase inhibitor, was initially evaluated in a phase 2a study of 14 patients with HDV [13]. Treatment of chronic HDV with oral lonafarnib significantly reduced virus levels, and the decline in virus replication significantly correlated with serum drug levels. In particular, the most effective dose was 200 mg given twice daily. Unfortunately, this dosage was aggravated with significant side effects. For this reason, additional trials were performed using lonafarnib 100 mg twice daily, either ritonavir-boosted or in combination with pegylated interferon. Both strategies were associated with a >2 log drop in HDV viremia, but a significant proportion of patients continued to experience adverse events [14].

Nucleic acid polymers (NAPs) are broad-spectrum antiviral agents whose antiviral activity in hepatitis B virus (HBV) infection is derived from their ability to block the release of the hepatitis B virus surface antigen (HBsAg). This pharmacological activity blocks HBsAg replenishment in the circulation, allowing host-mediated clearance. REP 2139 is a nucleic acid polymer that has been shown to clear HBsAg by blocking the release of subviral particles. This agent was evaluated to treat HDV infection in an uncontrolled phase 2 study [15]. Reported results in this pilot study were encouraging, with HDV suppression rates above 80% during treatment and maintained after treatment in more than 50% of patients. However, additional data to support these promising findings are necessarily needed.

In July 2020, the European Medicines Agency (EMA), Amsterdam, The Netherlands, issued a conditional marketing authorization for a new drug, bulevirtide (previously known as myrcludex-B), with the therapeutic indication for the treatment of chronic hepatitis Delta virus (HDV) infection in HDV-RNA positive adult patients with compensated liver disease. It was the first time a drug was specifically approved to treat Delta hepatitis.

Bulevirtide is an HDV entry inhibitor acting upon the sodium taurocholate co-transporting polypeptide (NTCP), a receptor shared by HBV and HDV, able to block cell entry of HBV and HDV. In an initial clinical study, bulevirtide was administered to 14 individuals, either as monotherapy (2 mg subcutaneously daily) or with pegylated IFN for 24 weeks [16]. Bulevirtide was well tolerated both as monotherapy and in combination with pegylated IFN. After 24 weeks of treatment, HDV-RNA decreased in all patients with chronic hepatitis B and D. Of the 14 patients who received bulevirtide, 13 experienced a >1 log10 reduction in HDV-RNA after 24 weeks of therapy. In addition, two of the seven patients became HDV-RNA negative in the monotherapy arm, compared with five of the seven patients who received combination therapy. However, hepatitis B surface antigen (HBsAg) levels remained unchanged. This initial study was the basis for two following phase 3 studies: Myr204 and Myr301. In the Myr204 study [17], investigators evaluated the safety and efficacy of bulevirtide administered subcutaneously at a dose of 2 or 10 mg daily in combination with pegylated interferon alfa-2a weekly relative to bulevirtide 10 mg monotherapy. Bulevirtide monotherapy and in combination with Peg-IFNα-2a was safe and well-tolerated through 24 weeks of therapy. Combination therapy and bulevirtide monotherapy resulted in high rates of HDV viral decline, well above 70%. In addition, bulevirtide monotherapy resulted in the highest rate of ALT normalization (64%). A combined response (>2 log drop in HDV RNA levels + ALT normalization) was obtained in 24–30% of patients in the combination arms and 50% of patients in the bulevirtide monotherapy arm. In the Myr301 [18], investigators evaluated the safety and efficacy of bulevirtide administered subcutaneously for a minimum of 48 weeks, at a dose of 2 or 10 mg daily, compared with no treatment. Patients enrolled to receive the EMA authorized prescription dose of 2 mg daily had a virologic response of 55%, a biochemical response of 53% and a combined response of 37%.

The efficacy of bulevirtide in chronic Delta hepatitis was confirmed in an open-label real-life French study: 2 mg subcutaneously daily induced on treatment virological and combined virological and biochemical response in 79% and 43%, respectively [19]. In addition, despite the EMA authorization being limited to patients with compensated liver disease, real-life experience has confirmed the safety and efficacy of bulevirtide monotherapy even in patients with decompensated liver disease [20]. Finally, and interestingly, the efficacy of bulevirtide does not seem to be influenced by different infecting HDV genotypes [21].

## 6. What Has 45 Years of Experience Taught Us?

After reviewing this long journey in managing Delta hepatitis, we have learned some firm lessons, and we are probably approaching a new era of better strategies to treat this severe liver condition (Table 1).

Interferon alfa therapy is probably not over, especially in the pegylated form. Peg-IFN treatment is still recommended by AASLD, EASL and APASL guidelines (even if it has never been approved for this indication by the FDA or the EMA). Peg-IFN (48–52 weeks) has proven effective in about 25% of patients, but a late relapse may occur.

Bulevirtide is the first and only approved (by the EMA, not yet by the FDA) drug for Delta hepatitis. The daily subcutaneous dose of 2 mg inhibits HDV replication and reduces ALT values in about 40% of patients. However, the efficacy of bulevirtide is likely to wane after withdrawal; therefore, long-term maintenance therapy probably should be prescribed.

NUCs, even if not effective against HDV replication, should probably be administered to HBV/HDV co-infected patients to prevent HBV flares.

Finally, in newly diagnosed patients, without decompensated disease and contraindications to interferon, a possible effective combination treatment could be Peg-IFN (48–52 weeks) plus bulevirtide (long-term) and eventually plus NUC (long-term) in order to control and prevent the severe outcome of this aggressive form of chronic hepatitis. The real-world experience will probably show us whether a new era has come for patients with Delta hepatitis.

## Figures and Tables

**Figure 1 jcm-11-01587-f001:**
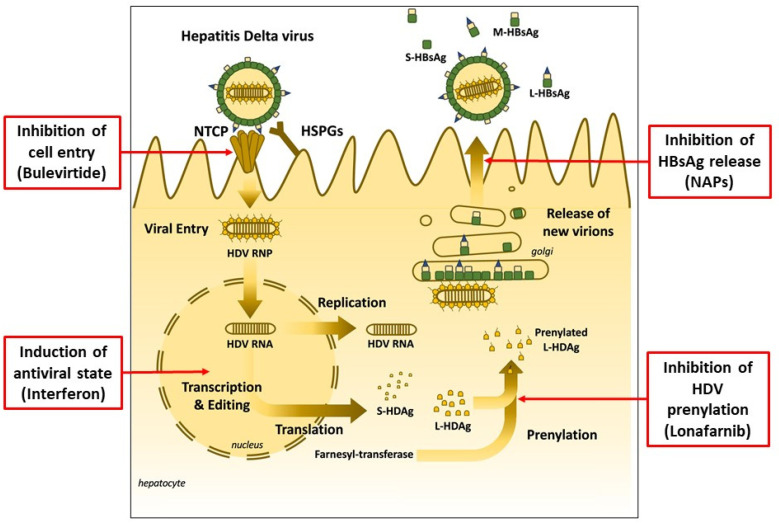
Hepatitis Delta Virus infection and therapeutic targets of currently available and experimental new drugs (adapted from *Liver Int.*
**2021**, *41* (Suppl. 1), 30–37, used with permission).

**Table 1 jcm-11-01587-t001:** Available and experimental drugs for the treatment of Delta hepatitis. Combination therapy trials are underway (Bulevirtide + Peg-IFN-α, Lonafarnib + Peg-IFN-α, NAPs + Peg-IFN-α).

Drug (Route)	Action	Dosage/Duration	Expected Response Rates (%)	Use in Decompensated Cirrhosis	Recommended by	Approved by
Peg-IFN-α (s.c.)	Immune modulation/enhancement	180 mcg weekly/12–18 months	23–43	No	AASLD, EASL, APASL	None
Bulevirtide (s.c.)	HDV cell entry inhibition	2 mg daily/long-term	37–55	No (recommended), Yes (real-life evidence)	NICE	EMA
Lonafarnib (oral)	HDV prenylation inhibition	50 mg + RTV bid/24–48 weeks	No data available	No data available	None	None
NAPs (REP 2139) (i.v.)	HBsAg release inhibition	500 mg/15 weeks	33	No data available	None	None

(Abbreviations: Peg-IFN: pegylated interferon, s.c.: subcutaneous, i.v.: intravenous, RTV: ritonavir, AASLD: American association for the study of liver diseases, EASL: European association for the study of the liver, APASL: Asian pacific association for the study of the liver, NICE: National institute for health and care excellence, EMA: European medicines agency).

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
