# Peer review of "Management of Delta Hepatitis 45 Years after the Discovery of HDV"

_jcm, 2022, doi:10.3390/jcm11061587_

Round 1
Reviewer 1 Report
Management of Delta hepatitis 45 years after the discovery of 2 HDV
This is an article in which the author performs a narrative review mainly on the treatment of HDV.
we suggest to the author:
1.Before starting with section 1. To make a section 0 with a brief introduction, justification and objectives of the review.
2. Include a figure of the HDV cycle with the different therapeutic targets.
3.To elaborate a table with the available and future drugs for HDV treatment with the main characteristics and indications for use.
Author Response
We thank the reviewer for the comments and helpful suggestions.
Suggestion 1. Before starting with section 1. To make a section 0 with a brief introduction, justification and objectives of the review.
Reply 1: An introductory section has been added to the manuscript.
Suggestion 2. Include a figure of the HDV cycle with the different therapeutic targets.
Reply 2: The figure has been included.
Suggestion 3. To elaborate a table with the available and future drugs for HDV treatment with the main characteristics and indications for use.
Reply 3: A table has been elaborated and added to the manuscript.
Reviewer 2 Report
This review is a detailed well-organized paper describing step by step all the problems and challenges in the therapy management of Delta hepatitis.
It is clearly pointed out that HBsAg clearance should be the ultimate goal to be achieved in order to have a long-term outcome with no liver-related problems.
It was revealed very well the role of each treatment approach and the progress made with the impact on both HDV replication and clinical outcome.
The author shows in a well-documented manner the complex aspects of Delta hepatitis, referring to the lack of testing standardization and response criteria, HBV elimination, transaminase flares, and poor response rates.
The conclusion after 45 years of experience in the management of HDV could be the therapeutic combinations between Peg-IFN (48-52 weeks) plus long-term Bulevirtide and, eventually, plus long-term nucleotide/nucleoside (NUC) analogs.
The message of the article is that more research is needed in order to be developed new drugs that will enable the patients to develop HBsAg seroconversion meaning also clearance of serum HDV RNA.
Author Response
We are very grateful to the reviewer for the comments, and no specific changes to the manuscript have been suggested.
Round 2
Reviewer 1 Report
In this second revised version the author has introduced the requested changes:
1.Make a section 0 with a brief introduction, justification and objectives of the review 2.
2. Include a figure of the HDV cycle with the different therapeutic targets. It would be necessary to know if you have permission to use this figure from another article. This is an aspect that I am not aware of and that the Editorial Committee should take into account.
Prepare a table with the available and future drugs for HDV treatment with the main characteristics and indications for use.
Author Response
Comment 1. Make a section 0 with a brief introduction, justification and objectives of the review 2.
Response: Done
Comment 2. Include a figure of the HDV cycle with the different therapeutic targets.
Response: Done
Comment: It would be necessary to know if you have permission to use this figure from another article. This is an aspect that I am not aware of and that the Editorial Committee should take into account.
Response: The adapted figure is based on a figure published in Liver International by John Wiley and Sons. John Wiley and Sons is a signatory to the STM Permissions Guidelines, which enable fellow signatory publishers to reuse up to 3 figures/tables free of charge. A grant of license has been obtained and it has been uploaded.
Comment 3. Prepare a table with the available and future drugs for HDV treatment with the main characteristics and indications for use.
Response: Done
